# A Microfluidic Concentration Gradient Maker with Tunable Concentration Profiles by Changing Feed Flow Rate Ratios

**DOI:** 10.3390/mi11030284

**Published:** 2020-03-10

**Authors:** Tao Zhang, Jiyu Meng, Shanshan Li, Chengzhuang Yu, Junwei Li, Chunyang Wei, Shijie Dai

**Affiliations:** 1Hebei Key Laboratory of Robotic Sensing and Human-robot interactions, School of Mechanical Engineering, Hebei University of Technology, Tianjin 300132, China; 18222962172@163.com (T.Z.); 201521202004@stu.hebut.edu.cn (J.M.); 201621202035@stu.hebut.edu.cn (C.Y.); 201621202078@stu.hebut.edu.cn (C.W.); 2National Key Laboratory of Reliability and Electrical Equipment, Hebei University of Technology, Tianjin 300130, China; 3Department of Computer Science and Electrical Engineering, Hebei University of Technology, Langfang 065000, China; junwei_li@hebut.edu.cn; 4Institute of Biophysics, School of Science, Hebei University of Technology, Tianjin 300401, China

**Keywords:** microfluidics, concentration gradient, nonlinear, feed flow rate ratios mixing

## Abstract

Microfluidic chips—in which chemical or biological fluid samples are mixed into linear or nonlinear concentration distribution profiles—have generated enormous enthusiasm of their ability to develop patterns for drug release and their potential toxicology applications. These microfluidic devices have untapped potential for varying concentration patterns by the use of one single device or by easy-to-operate procedures. To address this challenge, we developed a soft-lithography-fabricated microfluidic platform that enabled one single device to be used as a concentration maker, which could generate linear, bell-type, or even S-type concentration profiles by tuning the feed flow rate ratios of each independent inlet. Here, we present an FFRR (feed flow rate ratio) adjustment approach to generate tens of types of concentration gradient profiles with one single device. To demonstrate the advantages of this approach, we used a Christmas-tree-like microfluidic chip as the demo. Its performance was analyzed using numerical simulation models and experimental investigations, and it showed an excellent time response (~10 s). With on-demand flow rate ratios, the FFRR microfluidic device could be used for many lab-on-a-chip applications where flexible concentration profiles are required for analysis.

## 1. Introduction

With their advantages of low reagent consumption, large integration in small footprints, near-zero dead volume [1], and low-cost fabrications, polydimethylsiloxane (PDMS) microfluidic devices are increasing used for lab-on-a-chip applications. Most biological or chemical samples are composed of a mixture of several kinds of reagents, and thus on-chip mixing is an essential step in many applications [2]. With the help of external resources, such as electric signals [3], magnetic fields [4], acoustic energy [5], or light sources [6], microfluids can be mixed efficiently. In fact, the ability to generate stable linear [7] or nonlinear spatial chemical gradient profiles within microfluidics has been adapted extensively for analysis [8]. Past research studies have shown that a device that can generate spatially and temporally controlled gradients is regarded as a robust and powerful method to investigate migratory cells [9,10,11,12], drug screening [13,14], oil production screening from microalgae [15], nucleation and growth of crystals [16], and wound healing [17] under a variety of conditions.

Generally speaking, a linear concentration gradient generator can be easily set up by classic Christmas-tree microfluidic channel networks [8], convection-driven flow [18], or novel designs such as centrifugal microfluidics [19,20,21] or 3D-printing stereo networks [22]. To address a nonlinear concentration gradient profile, a range of different microfluidic mixing technologies have been developed. One of the earliest designs was a design for controlled diffusive mixing of reagents, which generated a range of shapes for the gradients, such as linear, parabolic, and periodic [8]. Then, another impressive device was developed to generate dynamic temporal and spatial concentration gradients using one single microfluidic device. By using this device, linear gradients with dynamically controlled characteristics (i.e., slope, baseline, or direction of the line), and nonlinear gradients with controlled nonlinearity [10] were illustrated.

In addition, a systematic investigation was carried out to demonstrate the universal approach [23] to generate stable gradients of any profiles by experimental and mathematical methods. This approach has been successfully used to mix many kinds of microfluids. In this work, it was shown that the concentration gradient profile could be controlled not only by the dynamic inputs but also by channel geometries. One more example, a structure-based approach, was presented to generate linear or nonlinear chemical gradients [24]. By varying the lengths of microchannel, the volume of dispensed sample solutions was dictated, and then complex gradient profiles were generated. Another principle of structure-based mixing or concentration gradient makers is topological mixing schemes [25], which split, rotate, and recombine the microflows to perform efficient mixing by diffusion. By the combination control of flow rate and chip geometries, an automated drug dilution platform [26,27] was presented recently. Microfluidic concentration gradient generators are among the most efficient devices employed for precise generation of various concentration gradients. They have been widely studied in various fields in recent decades [28,29,30]. The concentration distribution profile within the channel is determined by the length of the channel, the splitting ratio of the fluids, and also the numbers of inlets and outlets. Thus, it may be necessary to redesign a footprint if applied to new occasions.

In this work, we describe a device fabricated by soft lithography and consisting of a Christmas-tree network. As a multi-purpose concentration gradient maker, it has four independent inlets. In order to generate spatially and temporally controlled gradients, we present an FFRR (feed flow rate ratio) adjustment approach to generate nonlinear concentration gradient profiles. This device shows a fast response to make on-demand concentration gradient profiles within the microchannels. Also, complex nonlinear concentration distribution patterns were performed with one single microfluidic device by varying the feed-in flow rate of each inlet online.

Furthermore, the concentration profile of the device has unique geometric diversity characteristics. We successfully demonstrated the generation of bell-type and S-type concentration profiles by numerical and experimental approaches. A nonlinear concentration gradient could be created at sub-second temporal resolution, by feeding the device with different flow rates. Furthermore, it is not necessary to redesign footprints to generate a totally different concentration gradient. Actually, the only thing we need to do is tune the on-demand flow ratios of each inlet. This provides a method to generate symmetrical/asymmetrical, nonlinear, curvilinear flow patterns within one single microfluidic device. By online tuning of on-demand flow rate ratios, the FFRR microfluidic device could be used for many lab-on-a-chip applications where flexible concentration profiles are required for analysis.

## 2. Materials and Methods 

### 2.1. Chip Design and FFRR Strategy

As shown in Figure 1a, the Christmas-tree-type mixing unit includes four independent inlets, named 1, 2, 3, and 4, respectively, and one outlet. The diffusion length *L_d_* << *w*; here, *w* stands for the channel width. In order to perform a full mixing in the device, the channel length L should be long enough. Since the diffusion time *t_d_* = *L_d_*^2^/(2*D*), the lower limit of channel length to perform a completely mixing is around *L_min_* =*Qt_d_*/(*wh*), where *D*, *h*, and *Q* stand for the diffusion coefficient, channel height, and flow rate, respectively. In our design, it is assumed that *L* >> *L_min_* to make sure that all the mixing within the microchannels is completely developed. Moreover, in order to echo the concentration profiles of mixers with an even number of inlets, the layout of a microfluidic concentration gradient maker with five inlets and one single outlet is designed, as shown in Figure 1b. Here, we define these devices as type IV and type V, respectively. Then, we denote the status of these independent inlets by two matrixes, as in the following: 

(i) Array *C* stands for the state of their initial mass concentrations.

*C* = (*C*_1_, *C*_2_, *C*_3_, *C*_4_)/(*C*_1_, *C*_2_, *C*_3_, *C*_4_, *C*_5_);

(ii) Array *Q* stands for the state of their initial flow rate.

*Q* = (*Q*_1_, *Q*_2_, *Q*_3_, *Q*_4_)/(*Q*_1_, *Q*_2_, *Q*_3_, *Q*_4_, *Q*_5_);

It is straightforward to use the normalized mass concentration or flow rate as a label for the status of each inlet. The observation position of normalized concentration is shown as the black arrow in Figure 1. When each stream converges into a single channel before outlet, it shows a laminar flow state and almost no mixing. Therefore, in this straight channel, any position could be selected to analyze the concentration profiles. The normalized position is between 0 and 1, as shown in blue font in Figure 1. For example, *C*_i_ = 0 (i = 1, 2, 3, 4, 5) denotes that the initial mass concentration of inlet i is zero, i.e., the background buffer solution was feed into inlet i. Meanwhile, *C*_i_ = 1 (i = 1, 2, 3, 4, 5) denotes that the initial mass concentration of inlet i is one, i.e., the dye solution was feed into inlet i. The normalized flow rate is demonstrated as *k*_i_ = *Q*_i_/*Q*_in_, in which *Q*_in_ = ∑*Q*_i_. There are (2^4 or 5^−2) ways of feeding choices to mix the background buffer and dye solutions. Taking mirror/symmetrical into consideration, we take four typical modes to investigate the mass transfer and mixing performance of the type IV and type V devices, respectively, as shown in Table 1. Here, we set the status of inlet 1 as always the same as inlet 2 and inlet 4 as always the same as inlet 5 for the type V device. As such, this device can be considered as a special, multi-step, microfluidic concentration gradient maker device with three inlets.

### 2.2. Settings for Numerical Simulation

The mass transfer and concentration distribution at the outlet of the microfluidic device were mainly dependent on the flow rate percent and the initial concentration at each inlet. In order to investigate the mass transfer performance of the microfluidic concentration gradient maker, we first calculated the normalized concentration as a function of the normalized position at the outlet using a numerical model. In the numerical model, the Navier–Stokes equation (Laminar Flow module) and mass transfer equation (Transport of Diluted Species module) were solved by Comsol Multiphysics. The control equations and boundary conditions of the flow field and concentration field follow the settings of our previous work [3]. In the simulation model, the geometry of the model was the same as the real devices in the experimental sections. The width of the channel is *w* = 100 μm, the height is *h* = 40 μm, and the width of the gap is also 100 μm. Furthermore, the diffusion coefficient of the dye reagent was set as 1e^−12^ (m^2^·s^−1^). The density and viscosity of media were 1e^3^ (kg·m^−3^) and 1e^−3^ (Pa·s). The initial concentrations of the background media and dye were set as 0 and 1 mM, respectively.

### 2.3. Experimental Setup

The concentration gradient maker device was plasma-bonded by a polydimethylsiloxane (PDMS) channel and a glass slider. By using soft-lithography techniques, the PDMS channel was fabricated entirely by micromolding of Dowcorning Sylgard 184 and an elastomer. All the inlets of the microfluidic device were connected to syringe pumps (LSP01-3A, Longer, Baoding, China) using silicone tubes. Micrographs were acquired with a microscope (E100, Nikon, Tokyo, Japan) fitted with a digital camera (DS-Qi1Mc, Nikon, Tokyo, Japan). Mass transfer and concentration distribution within the device were visualized with red dyes dissolved in deionized (DI) water. Flow rate was set by the syringe pumps and measured by flow sensors (MFS5, Elveflow, Paris, France). The on-demand feed flow rate of DI water or dyes could be tuned by Longer syringe pump controllers. Images of mixing performance could be recorded by the digital camera in real time. In order to achieve a good observation of the final mixing performance, the outlet of the mixer must be within the field of view. Concentration distributions and profile plots are obtained by grayscale data across the outlet, as shown in Figure 1. From position 0 to position 1, we choose the grayscale along the “0–1” direction as the concentration profile across this line. For statistical purposes, three paralyzed lines and their concentration profile plots are analyzed. The average value of the concentration plots of the three lines can be a measurement indicator of the outlet.

## 3. Results and Discussion

### 3.1. Concentration Distribution Profiles of the Type IV Mixer

A series of numerical calculations are performed according to the flow rate and concentration matrix distribution from the above table. Figure 2 shows the simulation results of the concentration distribution at the outlet of the microfluidic device. Here, we began with the case of *C* = (0, 1, 1, 0) and *Q*_1_ = *Q*_3_, *Q*_2_ = *Q*_4_, as shown in Figure 2a. When k_1_ = k_3_ = (0.05, 0.15, 0.25, 0.35, 0.45), under different FFRR distribution parameters, the concentration distribution is significantly different. A highly nonlinear distribution law is presented. There are concentration inflection points at different locations with a relatively large concentration gradient range, and the concentration can reach a peak in every curve. Then, we considered the case of *C* = [0, 1, 1, 0] and *Q*_1_ = *Q*_4_, *Q*_2_ = *Q*_3_, as shown in Figure 2b. In this case, the distribution law of concentration gradient is quite different from that in the first case. The concentration gradient curve shows the trend of first rising and then falling, and reaches its maximum value at the normalized position of x = 0.5, when k_1_ = k_4_ = (0.05, 0.15, 0.25, 0.35, 0.45). Moreover, the pattern of these curves is axisymmetric on the x = 0.5 axis. However, the concentration gradient range of each curve is moderate. Figure 2c shows the case of *C* = (0, 1, 1, 1) and *Q*_1_ = *Q*_3_, *Q*_2_ = *Q*_4_. The trend of concentration gradient distribution is similar to Figure 2b; however, the curve is a monotonically decreasing function without an extreme value. Finally, Figure 2d shows the case of *C* = (0, 1, 1, 1) and *Q*_1_ = *Q*_4_, *Q*_2_ = *Q*_3_. The trend of concentration gradient distribution is similar to Figure 2a, but the difference is the poor symmetry. That is to say, this microfluidic device provides a simple, compact, and user-friendly approach to generate or maintain a customized dye molecular concentration gradient. All these results were obtained by adjusting the FFRR and concentrate matrix on the same microfluidic chip device.

The experimental verification of the reagent concentration distribution was performed by a set of tests. All the concentration profiles of the outlet were characterized indirectly using microscopic graphs. All the mass-transfer micrographs in this experimental section were taken by a digital camera. The microscopic graphs indicate the linear dependency of the grayscale on the mass distribution of dye. The concentration distributions, as shown in Figure 3a, provide the mass transfer performance of the microfluidic concentration gradient maker. The normalized concentration of dye varied with the normalized position across the outlet of the microchannel. Figure 3a shows the experimental results of the concentration distribution at the outlet of the microfluidic device. A case study of *C* = (0, 1, 1, 0) and *Q*_1_ = *Q*_3_, *Q*_2_ = *Q*_4_ is shown in Figure 3a. To investigate the effect of flow ratios among the inlets on the final concentration profiles, we recorded microscopic images of the mass distributions at the outlet and compared them with the numerical simulations, as shown in Figure 3b,c. The normalized concentration measurements and calculations in Figure 3d,e provide a semi-quantitative comparison of Figure 3b,c, respectively. The horizontal axis and vertical axis also represent the normalized position and normalized concentration, respectively. The concentration profiles shown in Figure 3d are the average results of five experimental groups. Data plots with error bars are included in the Appendix A. The results show that the maximum error is 15.1% and most of the errors are within 10%, which shows the good concentration reproducibility generated using this device. In both cases, the concentration profiles behave like a bell-type curve. In particular, during image processing, in order to eliminate the errors of light or noise between images, the gray values are all normalized between 0 and 1, as shown in Figure 3d(i), and after smoothing, in Figure 3d(ii). 

In order to compare the experimental results with numerical simulation data, the concentration distribution data of Figure 3c were normalized, as shown in Figure 3e. From Figure 3d,e, the experimental and simulation data in blue curves, which stand for the cases of k_1_ = k_3_ = 0.25, all behave like a bell-type curve. However, for other cases, the experimental results are not very consistent with the simulation. The concentration profile plots from experiments are symmetric bell-like curves, while the plots of simulation data show asymmetric behavior. One possible reason for this is that the boundary conditions of the flow rate at the channel wall in the numerical simulation model are set as zero. The initial normalized concentration is set as zero (DI water) or one (dye). If the diffusion coefficient in this model is not large enough, the diffusion at the channel wall may remain at its initial value, as demonstrated in Figure 3e. On the other hand, the feed flows are generated by external syringe pumps, which may provide a vortex or disturbance caused by flow fluctuation. The vortex leads to a stronger diffusion, especially at the channel wall. Thus, the starting points of concentration profile plots are not exactly the same as their initial values (zero or one).

### 3.2. Concentration Distribution Profiles of the Type V Mixer

Motivated by the type IV concentration profile, whose concentration gradient curve shows good symmetry and regularity in experimental studies, we then thoroughly change the number of inlets to an odd number of type V concentration profiles, as shown in Figure 4, to find a richer concentration gradient distribution.

As shown in Figure 4a–d, the normalized concentration as a function of normalized position at the outlet of the microfluidic device was plotted. Here, the concentration matrix for Figure 4a,b is *C* = (0, 0, 0, 1, 1), and the concentration matrix for Figure 4c,d is *C* = (0, 0, 1, 0, 0). Compared with the type IV concentration gradient maker, the concentration gradient form generated by the type V device is more abundant and various. From Figure 4a, the normalized concentration is monotone, increasing against the normalized position, when k_1,2_ = k_3_ = 0.1, 0.2, 0.3, 0.4. With the increase in k, the initial value of the concentration gradient at the outlet decreases gradually, and the gap is large. When k_1,2_ = k_3_ = 0.5, only buffer solutions are fed, so the concentration is 0. However, when k_1,2_ = k_4,5_, with the increase in k, the concentration increases and the initial value is similar, but the final values present seemingly uniform differences, as shown in Figure 4b. Instead of going all the way to an allowed maximum value of 1, here, the normalized concentration shown in Figure 4c eventually reached a peak and decreased once more. Similarly, when the concentrations reached the maximum values defined by diffusion and convection, their quantities decreased with the channel width position, as shown in Figure 4c. With the increase in k, the position of the peak value moves to the right. The concentration profile shown in Figure 4d is similar to that shown in Figure 2b. However, the difference is that when k_1,2_ = k_4,5_ = 0.2 and 0.3, the concentration gradient has a relatively large dynamic range.

One of the concentration profile patterns was used to carry out experimental verification. Figure 5 demonstrates the numerical and experimental results of the S-type concentration distribution profiles, with the condition of *C* = (0, 0, 0, 1, 1) and *Q*_1,2_ = *Q*_4,5_. Figure 5a shows the experimental results of the concentration distribution at the outlet of the type V device. To investigate the effect of flow ratios among the inlets on the final concentration profiles, we recorded microscopic images of the mass distributions at the outlet and compared them with numerical simulations, as shown in Figure 5b,c. The normalized concentration measurements and calculations in Figure 5d,e provide a semi-quantitative comparison of Figure 5b,c, respectively. The data processing method in Figure 5d,e is the same as that in Figure 3d,e. Data plots with error bars are included in the Appendix A. The results show that the maximum error is 13.7% and most of the errors are within 10%, which shows the good concentration reproducibility generated using this device. All concentration curves fall slightly first, then rise to a peak and gradually reach the saturation value. Compared with the simulation data, the experimental results have the characteristic of small differences between each curve. A possible reason for this is background noise, as discussed in the previous subsections. Compared with the numerical curves, the experimental results show a compact concentration profile along with the normalized position. A possible reason for this is that the diffusion coefficient in the numerical model is larger than that of the dye. As a result, the diffusion process in the experiments is not as fast as the calculated results. 

A physical explanation of microfluidic mass transfer can be interpreted by the Navier–Stokes equation for incompressible laminar flow and by the mass transfer equation for diluted species, as mentioned in Section 2.2. However, this approach can only provide numerical results for concentration profiles. It is still unable to provide a straightforward closed-form solution to discuss why the specific patterns of the concentration profile are formed in terms of fluid dynamics. Previous studies have attempted to give a general expression to describe concentration profiles by using recursive formulas [8]. Here, we provide a polynomial of nth order ($f\left(x\right)=a_{0}+a_{1}\times x+a_{2}\times x^{2}+\ldots +a_{n}\times x^{n}$) to describe the concentration profile of specific mixing networks. Here, $a_{n},\;x,\;\mathrm{and}\;n$ represent the fitting coefficient, variable, and order of the polynomial, respectively. This provides an empirical formula for a Christmas-tree microfluidic network with three inlets. Although this method is useful for biological applications, it is a mathematical rather than physical explanation. 

In future work, we may try to give a straightforward closed-form expression for concentration profiles of typical feed flow conditions. Also, we may update the diffusion coefficient of the dye to make the calculated curve agree with the experimental results. This also provides a potential approach to estimate the diffusion coefficient of unknown reagents.

## 4. Conclusions

We have demonstrated a microfluidic concentration gradient maker with four or five independent inlets and Christmas-tree networks to generate nonlinear concentration gradient profiles. Experimental investigation and numerical proofs were also performed. In order to obtain flexible concentration distribution profiles, the feed flow rate ratio (FFRR) needs to be well controlled. By varying the FFRRs, the technique was characterized as having two advantages: (1) An on-demand concentration gradient profile can be easily obtained by varying the flow rate of each independent inlet. (2) Linear or nonlinear concentration profiles, including bell-type and S-type profiles, can be realized by one single microfluidic device, and thus it is not necessary to renew the footprint for new occasions. The method shows promise for rapid on-demand mixing in lab-on-a-chip applications.

## Figures and Tables

**Figure 1 micromachines-11-00284-f001:**
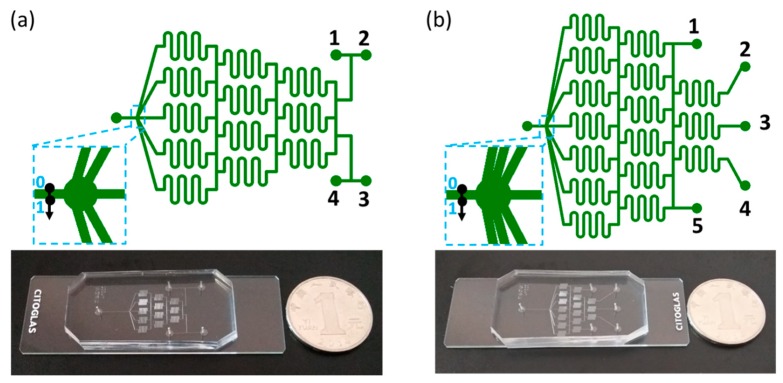
Schematic of the type IV (**a**) and type V (**b**) concentration gradient makers used in this work, and the corresponding experimental microfluidic chips fabricated with glass-polydimethylsiloxane (PDMS) by soft lithography and plasma bonding. Here, 1, 2, 3, 4, and 5 denote the five inlets.

**Figure 2 micromachines-11-00284-f002:**
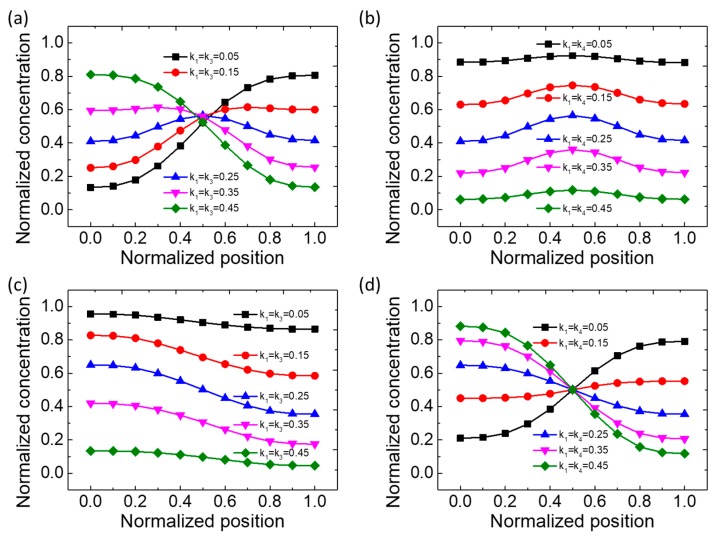
Numerical simulation results of normalized concentration as a function of normalized position at the outlet for the type IV microfluidic device. (**a**–**d**) show the cases of *C* = (0, 1, 1, 0) and *Q*_1_ = *Q*_3_, *Q*_2_ = *Q*_4_; *C* = (0, 1, 1, 0) and *Q*_1_ = *Q*_4_, *Q*_2_ = *Q*_3_; *C* = (0, 1, 0, 1) and *Q*_1_ = *Q*_3_, *Q*_2_ = *Q*_4_; and *C* = (0, 1, 0, 1) and *Q*_1_ = *Q*_4_, *Q*_2_ = *Q*_3_, respectively.

**Figure 3 micromachines-11-00284-f003:**
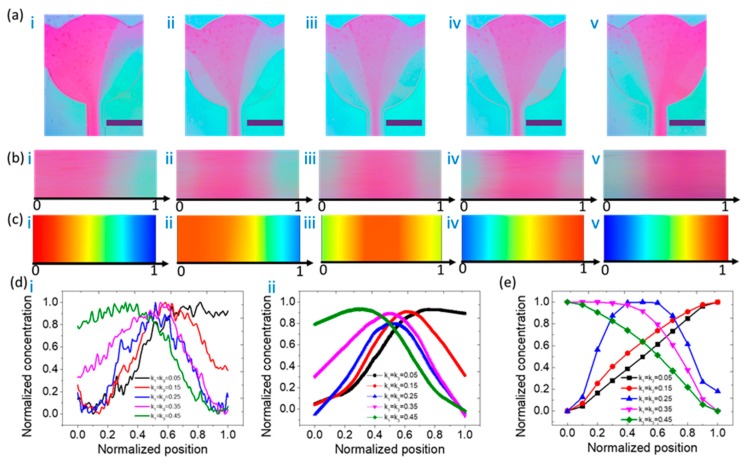
(**a**) Experimental results of the concentration distribution at the outlet of the microfluidic device. From left to right, i denotes the case of k_1_ = k_3_ = 0.05; ii denotes the case of k_1_ = k_3_ = 0.15; iii denotes the case of k_1_ = k_3_ = 0.25; iv denotes the case of k_1_ = k_3_ = 0.35; and v denotes the case of k_1_ = k_3_ = 0.45. The scale bar is 200 μm. The concentration matrix is *C* = (0, 1, 1, 0). (**b**) Partial enlarged view of the mass concentration of the outlet corresponding to (**a**). (**c**) Numerical simulation results of the concentration distribution at the outlet of the microfluidic device. From left to right, the flow ratios are shown in the cases of i to v. (**d**) Experimental results of normalized concentration as a function of normalized position at the outlet of the microfluidic device, whose pattern is like a bell-type curve. A smoothing process technique was adopted from i to ii. (**e**) Numerical simulation results of normalized concentration as a function of normalized position at the outlet of the microfluidic device.

**Figure 4 micromachines-11-00284-f004:**
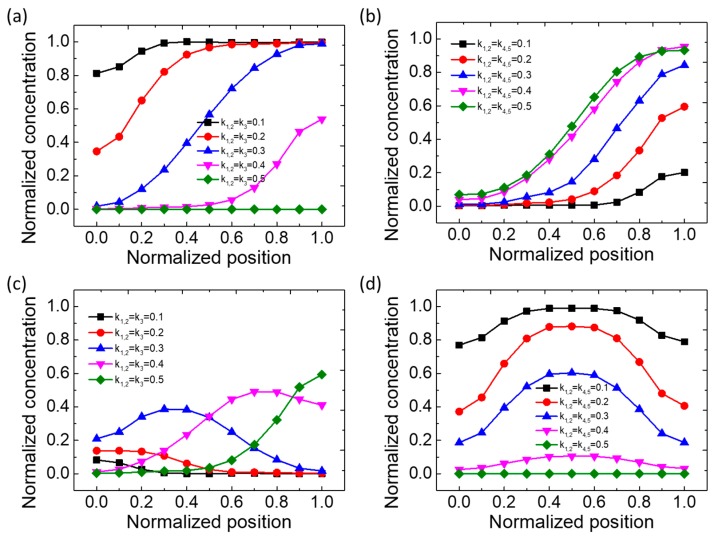
Numerical simulation results of normalized concentration as a function of normalized position at the outlet of the type V microfluidic device. (**a**–**d**) show the cases of *C*= (0, 0, 0, 1, 1) and *Q*_1,2_ = *Q*_3_, *Q*_4,5_; *C* = (0, 0, 0, 1, 1) and *Q*_1,2_ = *Q*_4,5_, *Q*_3_; *C* = (0, 0, 1, 0, 0) and *Q*_1,2_ = *Q*_3_, *Q*_4,5_; and *C* = (0, 0, 1, 0, 0) and *Q*_1,2_ = *Q*_4,5_, *Q*_3_, respectively.

**Figure 5 micromachines-11-00284-f005:**
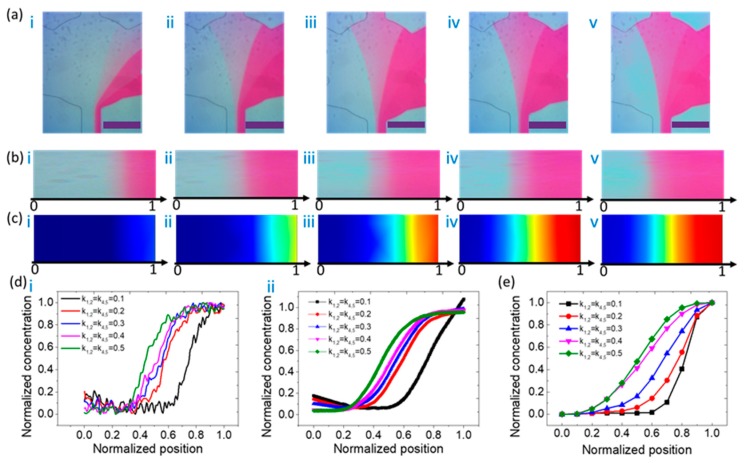
(**a**) Experimental results of the concentration distribution at the outlet of the microfluidic device. From left to right, i denotes the case of k_1,2_ = k_4,5_ = 0.1; ii denotes the case of k_1,2_ = k_4,5_ = 0.2; iii denotes the case of k_1,2_ = k_4,5_ = 0.3; iv denotes the case of k_1,2_ = k_4,5_ = 0.4; and v denotes the case of k_1,2_ = k_4,5_ = 0.5. The scale bar is 200 μm. The concentration matrix is *C* = (0, 0, 0, 1, 1). (**b**) Partial enlarged view of the mass concentration of the outlet corresponding to (**a**). (**c**) Numerical simulation results of the concentration distribution at the outlet of the microfluidic device. From left to right, the flow ratios are shown in the cases of i to v. (**d**) Experimental results of normalized concentration as a function of normalized position at the outlet of the microfluidic device, whose pattern is like an S-type curve. A smoothing process technique was adopted from i to ii. (**e**) Numerical simulation results of normalized concentration as a function of normalized position at the outlet of the microfluidic device.

**Table 1 micromachines-11-00284-t001:** Four typical statuses of type IV and type V concentration gradient maker devices.

Type	*C* _1_	*C* _2_	*C* _3_	*C* _4_	*C* _5_	Remarks
Type IV	buffer	dye	dye	buffer		Q_1_ = Q_3_, Q_2_ = Q_4_
	buffer	dye	dye	buffer		Q_1_ = Q_4_, Q_2_ = Q_3_
	buffer	dye	buffer	dye		Q_1_ = Q_3_, Q_2_ = Q_4_
	buffer	dye	buffer	dye		Q_1_ = Q_4_, Q_2_ = Q_3_
Type V	buffer	buffer	buffer	dye	dye	Q_1,2_ = Q_3_, Q_4,5_
	buffer	buffer	buffer	dye	dye	Q_1,2_ = Q_4,5_, Q_3_
	buffer	buffer	dye	buffer	buffer	Q_1,2_ = Q_3_, Q_4,5_
	buffer	buffer	dye	buffer	buffer	Q_1,2_ = Q_4,5_, Q_3_

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
