# Peer review of "A Microfluidic Concentration Gradient Maker with Tunable Concentration Profiles by Changing Feed Flow Rate Ratios"

_micromachines, 2020, doi:10.3390/mi11030284_

Round 1

Reviewer 1 Report

This manuscript suggests an on-chip mixing platform which enables to adjust concentration profiles with a passive method. The authors manipulate the concentration profiles by tuning the location of dye inlets and the flow rate ratio of each inlet. This method is interesting because it can reduce trial and error in finding the suitable conditions for certain patterns of concentration profile. In addition, the authors conducted both numerical simulation and experimental investigation to stress the accuracy of the research. However, because of a number of critical shortcomings, I determined that this manuscript is not appropriate to be published in Micromachines.

1. Introduction,

1-1. The authors couldn’t stress the importance of on-chip mixing. The authors suggested only one specific application (controlling gradients of chemotactic factors for investigation of migratory cells depending on conditions). A few more applications need to be added to emphasize the importance of on-chip mixing.

1-2. It would be great if the authors could explain that the major novelty of this research could help solve the issue of a specific application.

1-3. Except for Ref.18, most of the references regarding previous studies on on-chip mixing with microfluidics were studied over a decade ago. It can lead misunderstanding to others who are not familiar with on-chip mixing (for example, they can think that on-chip mixing has not been widely studied recently and is not an important field). More recent studies should be added in the introduction. It would also be great if authors could compare the pros and cons of this research and recent previous studies.

2. Material and Method,

2-1. The description that how numerical simulation was conducted (from line 132 to 137) should move to the material and method section.

2-2. Experimental information is insufficient. For example, the detailed information of the camera should be added in this section (including the company name). Furthermore, authors need to describe how to measure, calculate, and normalize concentration profiles from the experimental results.

3. Results and discussion

3-1. One of the major issues in this manuscript is that the physical explanation is insufficient. The authors need to add discussions on why the specific patterns of concentration profile are formed in terms of fluid dynamics. If the authors couldn’t figure it out, authors should at least present the tendencies of concentration profile formation depending on conditions. It could be a guideline to others who want to use this channel for on-chip mixing in the future.

3-2. A few explanations are in the manuscript about why some of the experimental results are inconsistent with the simulation. Authors mention that “buffer solution in the inlet 4 will not be thoroughly mixed with the dye solution in the inlet 3”, and this explanation is about the results from the 4 type mixer. But given the structure of the 4 type mixer in Figure 1, the channel shows a symmetric structure. I don’t agree that this situation could happen only on one side of the symmetric channel.

3-3. The experimental results are insufficient. The authors used two main parameters in this research: the flow rate ratio of each inlet and the location of dye inlet. But authors didn’t use the location of the dye inlet as a parameter in the experiment.

3-4. It might be great if authors could show a video about changing concentration profiles in real-time as the flow rate ratio is adjusted.

4. Figures

4-1. The authors used the expression “normalized position”. It might be great if authors insert a line scan into the channel schematic to mark the exact position corresponding to normalize position 0 and 1.

4-2. The description for each image is insufficient in each caption. Even authors explain the details in the manuscript, some information should be added to the captions. For example, what were the conditions about the location of dye inlets in Figure 2 (a), (b) and Figure 5 (a), (b). Furthermore, what was the main difference between i and ii in Figure 2(d) and 5(d).

4-3. The authors mentioned the specific graph in Figure 3 and Figure 5 as the curve with the solid square marker. I could easily find the graph in Fig (e), but it was difficult to find it in Fig (d). So it would be better to mention each graph using its color.

4-4. Authors suggest new words like spoon type and s-type, while authors are explaining graphs, but the spoon type, in particular, is hard to agree. In addition, even the authors mentioned the specific graph as the spoon type in the manuscript, but the authors mentioned the same graph as bell-type in the caption. Word coincidence is required.

5. Writing

5-1. Authors used some sentences like ‘what’s more’ and ‘what’s important’. I feel that these sentences are quite awkward.

Reviewer 2 Report

The authors described a microfluidic device designed to generate both linear and non-linear concentration gradients using the feed flow rate ratio changing mechanism. Overall the manuscript was well prepared, but improvement in the writing quality is expected. 

It was not clear about the definition for the normalized position, and thus unclear about the concentration that could be generated at each downstream channels before the final outlet. Please indicate the normalized position on Figure 1. 

Besides, the authors declared their device to generate stable concentration gradient. Please also provide results regarding the concentration reproducibility generated using this device.

Round 2

Reviewer 1 Report

The authors kindly responded to most of my requests. But I have one more request.

1.       In response to comment 3-1, the authors suggested a recursive formula to describe the concentration profile of specific mixing networks. Please insert the meaning of each parameter (a, x, n) in the case of this mixing platform.
